# Metabolic and Proteomic Profiling of Coronary Microvascular Dysfunction: Insights from Rat Models

**DOI:** 10.3390/biom14101305

**Published:** 2024-10-16

**Authors:** Yan Lu, Yuying Wang, Qiqi Xin, Rong Yuan, Keji Chen, Jianfeng Chu, Weihong Cong

**Affiliations:** 1Laboratory of Cardiovascular Diseases, Xiyuan Hospital, China Academy of Chinese Medical Sciences, Beijing 100091, China; yan_lu1123@126.com (Y.L.); wyybrave1@163.com (Y.W.); xinqiqiyuan@126.com (Q.X.); kjchenvip@163.com (K.C.); 2National Clinical Research Center for Chinese Medicine Cardiology, Beijing 100091, China; 3Academy of Integrative Medicine, Fujian University of Traditional Chinese Medicine, Fuzhou 350122, China

**Keywords:** coronary microvascular dysfunction, proteomics, metabolomics, machine learning, diagnostic marker

## Abstract

Coronary microvascular dysfunction (CMD) represents a principal etiological factor in ischemic heart disease. Nonetheless, a considerable subset of CMD patients experiences diagnostic delays attributable to the inadequacy of current diagnostic methodologies; which in turn results in deferred therapeutic interventions and elevated mortality rates. This study seeks to elucidate the distinct metabolic profile associated with CMD in rat models and to identify specific diagnostic markers that could enhance the diagnostic accuracy for CMD. In this study, 18 Wistar rats were randomly allocated into two groups: the sham group and the CMD group. The CMD group received injections of embolic microspheres into the left ventricle to establish a CMD model. Subsequently, non-targeted metabolomics and acetylated proteomics analyses were conducted. Machine-learning techniques were employed to identify the co-diagnostic markers of the disease. This study identified 53 key proteins through differential expression proteins (DEPs) and modular proteins analysis. Subsequently, four core proteins (Emc1; Ank1; Fbln2; and Hp) were determined as diagnostic markers for CMD using lasso regression, support vector machine, and random forest methodologies. Receiver operating characteristic curve analysis further demonstrated robust diagnostic performance. Gene ontology and kyoto encyclopedia of genes and genome enrichment analyses indicated that the DEPs were predominantly associated with metabolic pathways. Ultimately, the integrative analysis of proteomics and metabolomics suggested that the central metabolic mechanism underlying CMD pathogenesis may be linked to the tricarboxylic acid cycle. This study revealed specific changes in the proteomic and metabolic profiles of CMD rats and identified four diagnostic markers, which are proteins and metabolites that could be potential diagnostic biomarkers for CMD.

## 1. Introduction

Ischemic heart disease (IHD) is a leading cause of morbidity and mortality, affecting 112 million people worldwide in 2015 [1]. Coronary microvascular dysfunction (CMD) is one of the leading causes of IHD, accounting for approximately two-thirds of clinical conditions presenting with signs and symptoms of myocardial ischemia without obstructive coronary artery disease, including microvascular angina and myocardial infarction with non-obstructive coronary artery disease [2,3]. Despite CMD having a high prevalence in IHD and being associated with poor outcomes, the diagnosis of CMD remains challenging and its underlying mechanisms remain poorly understood [4]. Therefore, there is an urgent need to accurately define CMD, reveal its complex molecular features, and establish accurate detection techniques with multiple molecular markers to achieve early diagnosis and early warning, and to ensure that patients can receive accurate and effective interventions in a timely manner [5,6].

Invasive provocative testing in patients with angina and suspected CMD was found to reduce healthcare costs and improve quality of life [7,8]. Despite this finding, invasive testing remains underutilized. The introduction of widely used thermodilution systems has increased the accessibility of invasive testing for CMD, allowing measurement of coronary flow reserve (CFR) and index of microcirculatory resistance (IMR), but has also brought into question the optimal threshold for the diagnosis of CFR and increased the population of patients with discordant CFR and microcirculatory resistance indices [9]. The functional vasoreactivity aspect of CMD is equally complex and reflects an interplay of impaired, disrupted endothelial signaling, and altered metabolic responses. The comprehensive diagnosis of CMD is based on an in-depth understanding of coronary microvascular pathophysiology, which is determined by a combination of invasive and noninvasive diagnostic modalities. This diagnostic synergy is essential to differentiate CMD from other heterogeneous cardiovascular diseases, enabling the initiation of tailored therapeutic interventions [10]. With the development of multi-omics detection and analysis technology, the establishment of an accurate detection system for multiple molecular markers for early diagnosis, early warning and prediction of CMD based on evidence-based medicine and big multi-omics data would be a strategic breakthrough [11]. In this study, we employ a combination of proteomics and metabolomics, augmented by machine-learning techniques, to identify biomarkers and evaluate their diagnostic utility in CMD, which can be used for the early diagnosis of diseases.

## 2. Methods

### 2.1. Animals

Six-week-old male Wistar rats were used in this study and were obtained from Beijing Vital River Laboratory Animal Technology Co., Ltd. (Beijing, China). The rats were housed in groups under controlled conditions of light (12 h photoperiod) and temperature (22 °C), with unrestricted access to food and water. The animal experimental protocol of this study was approved by the Ethics Review Committee of Animal Experimentation of Xiyuan Hospital, China Academy of Chinese Medical Sciences (protocol code 2024XLC052-2 and date of 2 August 2024). All animal experiments followed the “ARRIVE (Animal Research: Reporting of In Vivo Experiments) guidelines”.

### 2.2. Rats CMD Model Establishment

In this study, 18 Wistar rats were randomly divided into 2 groups: sham and CMD, with nine rats in each group. After one week of acclimatization feeding, rats were anesthetized by intraperitoneal injection of 30–40 mg/kg pentobarbital hydrochloride in the CMD group, followed by tracheal intubation and insertion of a small animal respirator to assist with respiration. A longitudinal incision was made in the thorax between the third and fourth intercostal spaces to fully expose the heart and aorta. Subsequently, the ascending aorta was isolated, and 0.2 mL of embolized microspheres (each with a diameter of 42–120 µm, containing approximately 1000 microspheres) were rapidly injected from the apical portion directly into the left ventricle. Concurrently, the ascending aorta was clamped shut with hemostatic forceps for 10 s. After the heartbeat was stabilized, the pleural cavity of each animal was closed layer by layer with sutures. Once respiration was stabilized, the rats were extubated and then injected intraperitoneally with 800,000 units of penicillin, followed by regular postoperative feeding. A similar surgical procedure was performed in the sham group, except for an injection of 0.2 mL of saline without microspheres was injected.

### 2.3. Transthoracic Echocardiography

Echocardiographic measurements were recorded 4 weeks after surgery. Rats were anesthetized with 1% isoflurane concentration. The rats were gently immobilized on a warming pad in a supine position and the precordium was shaved. Transthoracic echocardiography was performed by use of a Vevo^®^ 2100 high-resolution small animal ultrasound imaging system. The left ventricular M-mode echocardiogram was obtained by scanning the left thoracic 3rd and 4th intercostal spaces of the rats, with the probe aimed at the left thorax and intercepting standard two-dimensional short-axis views.

### 2.4. Coronary Flow Reserve Testing

After obtaining the left ventricular long-axis section, rats were anesthetized by 1% isoflurane concentration. Coronary blood flow velocity signals were detected with a rat Doppler probe, with the detection site in the second to third intercostal space on the left side of the rat chest. The position was adjusted using a micropositioner to optimize the blood flow display signal on the display. Baseline data were recorded. Subsequently, the isoflurane concentration was increased to 2.5%. After 3 min of stabilization, the coronary blood flow velocity was significantly increased. After the waveform was stabilized, the data after coronary dilatation were recorded and saved. The data were analyzed and calculated by Doppler signal processing software. Coronary flow reserve (CFR) is the ratio of blood flow velocity in the dilated state of coronary artery to that in the resting state (CFR = V_max_/V_min_).

### 2.5. Heart Collection and Histological Analysis

At the study endpoint (week 4 after CMD), rats were anaesthetised as detailed in the description of surgical induction. The abdominal cavity was opened and the abdominal aorta was exsanguinated until the animal died. After reopening the thoracic cavity, the heart was removed and was washed of blood by saline perfusion. The hearts were split for subsequent experiments. Specifically, rat heart tissues were collected and immediately fixed in 4% paraformaldehyde for 24 h. Following paraffin embedding, the samples were cut into 4 μM slices. Slices were subject to deparaffinization and dehydration before H&E staining (Servicebio, Wuhan, China) of these tissues. Images were captured by a light microscope (Nikon, Minato City, Japan).

### 2.6. Assessment of Myocardial Ultrastructure by Transmission Electron Microscopy

After 4 weeks of CMD, tissue samples from the left ventricle of rats in each group were fixed in 2.5% glutaraldehyde for 24 h at 4 °C. For transmission electron microscopy analysis, the tissue samples were initially washed in phosphate buffer and subsequently fixed in 1% osmium tetroxide (Ted Pella Inc., Redding, CA, USA) at room temperature for 2 h. Subsequently, the samples were progressively dehydrated in 30%, 50%, 70%, 80%, 95%, and pure ethanol. Subsequently, the samples underwent two washes with pure acetone prior to the embedding process. The embedding procedure involved the gradual permeation of the samples with a series of acetone and 812 embedding agent mixtures in the ratios of 1:1 and 1:2, followed by immersion in pure 812 embedding agent. This process was conducted at 37 °C overnight. The samples were then polymerised at 60 °C for 48 h. Resin blocks were sliced into 60–80 nm ultrathin sections by an ultrathin slicer and then placed on a copper grid. Tissue sections were stained with 2% uranyl acetate saturated alcoholic solution in the absence of light for 8 min, followed by 2.6% lead citrate solution in the absence of carbon dioxide for 8 min. Images were acquired using an HT7800 transmission electron microscope (HITACHI, Chiyoda City, Japan).

### 2.7. Serological Tests

Rat serum creatine kinase MB isoenzyme (CK-MB) activity was determined using a rat CK-MB ELISA kit. Rat serum cardiac troponin T (cTnT) activity was determined using a rat cTnT enzyme-linked immunosorbent assay (ELISA) kit. All of the reagents above were purchased from Inselisa (Beijing, China), and the tests were performed according to the manufacturer’s instructions.

### 2.8. Sample Preparation Analysis for Proteomics

Myocardial tissue was ground individually in liquid nitrogen and lysed with lysis buffer. The samples were centrifuged at 25,000× *g* for 15 min to remove the precipitate. The supernatant was reduced with 10 mM DTT for 1 h, followed by IAM in the dark for 45 min. The supernatant was treated with 4 times the volume of precooled acetone and the precipitate was collected by centrifugation at 25,000× *g* for 15 min. Proteins were quantified using the bovine serum albumin assay, and the integrity was determined by electrophoresis. Peptide preparations were treated with trypsin at 37 °C for 4 h. The eluents were lyophilized, dissolved in binding buffer and loaded onto anti-PTM-agarose to enrich for acetylated peptides. After several washes, the eluate was collected, desalted and lyophilised. LC–MS/MS detection was performed using a timesTOF Pro spectrometer (Bruker, Billerica, MA, USA).

### 2.9. Protein Identification and Analysis

Identification was completed using the MaxQuant integrated andromeda engine. Proteins and modification sites were further filtered at less than 1% false discovery rate (FDR), while significant modifications were obtained. Protein expression was performed using the label-free quantification module. Proteins sharing the same set of peptides were automatically grouped together. Potentially contaminating sequences, reverse sequences, and proteins only “identified by site” were removed, and proteins with at least 2 unique peptides were identified. Furthermore, proteins with at least 2 valid values from one group were retained. We performed a principal component analysis (PCA) on the data, following standardization procedures. Differentially expressed proteins (DEPs) were analysed using the limma package. Significantly expressed proteins were defined as those with a fold change (FC) > 1.2 and *p*-value < 0.05. Then, the results were visualized in volcano plots and heat map plots which were generated with the R packages ggvolcano and pheatmap, respectively. The enrichment of DEPs in the kyoto encyclopedia of genes and genomes (KEGG) pathway was analysed using the clusterProfiler package. Pathways were considered to be significantly enriched when the *p*-value for the enriched pathway was <0.05.

### 2.10. Protein Co-Expression Network Analysis

As a systematic biological approach, weighted gene co-expression network analysis (WGCNA) was used to reveal patterns of gene associations between different samples and to detect candidate biomarkers or therapeutic targets based on the interconnections between sets of genes as well as the associations between sets of genes and phenotypes [12,13]. Correlations between modules and clinical characteristics are calculated to identify modules that are highly correlated with protein modules of the clinical phenotype. The function WGCNA: ‘blockwiseModules()’ was applied to construct the co-expression network with the following settings, a soft threshold power β = 10 (calculated based on the scaled-free topology model parameters), minModuleSize = 30, mergeCutHeight = 0.25. All other parameters were maintained at their default settings. Following the acquisition of the modules, the distinct module eigengenes were derived from the first principal component of the module expression profiles, while the module–trait relationships were evaluated based on the association of module eigengenes with clinical features. Modules exhibiting the most significant positive and negative correlations in the module–trait relationships were then identified. Then, the MM and GS scores in the modules were assessed to determine the module significance.

### 2.11. Machine-Learning Prediction

In order to identify candidate diagnostic markers, 3 machine-learning algorithms were employed to analyze key proteins and differential metabolites: lasso regression, random forest, and support vector machine (SVM). These methods were applied by R and bioconductor packages ‘glmnet’, ‘random forest’, ‘e1071’. The receiver operating characteristic (ROC) curves and area under the curve (AUC) were calculated by the R package ‘pROC’. Following this, the expression levels of the identified biomarkers were examined by using the ELISA method validated to assess their potential as diagnostic biomarkers.

### 2.12. Non-Targeted Metabolome Analysis

For the analysis of myocardial metabolites, a Waters 2777C Ultra Performance Liquid Chromatography (UPLC) system (Waters, Milford, MA, USA) and a Q Exactive HF high-resolution mass spectrometer (Thermo Fisher Scientific, Waltham, MA, USA) were employed. Metabolites were separated using an ACQUITY UPLC BEH C18 column (2.1 × 100 mm, 1.7 μm particles), which was maintained at a column temperature of 45 °C. The injection volume was 5 μL, and the flow rate was 0.35 mL/min. Mobile phase A was aqueous solution containing 0.1% formic acid, and mobile phase B was methanol containing 0.1% formic acid. The linear gradient began with an increase from 1% to 2% B in 1 min, followed by a rise from 2% to 98% B in 9 min. Subsequently, the gradient was reduced to 2% over 12.1 min and maintained for 2.9 min. The mass spectrometry scanning mass-to-core ratio ranged from 70 to 1050 *m*/*z* with a first-order resolution of 120,000 and a maximum injection time of 0.1 s.

The second-order resolution was set to 30,000, with a maximum injection time of 50 ms. The fragmentation energies were adjusted to 20, 40, and 60 eV to obtain accurate masses of product ions. Ionization of the eluent was performed using an ESI source at a sheath gas flow rate of 40 L/h and an auxiliary gas flow rate of 10 L/h. The spray voltages were 3.8 V for the positive ion mode and 3.2 V for the negative ion mode, and the temperature of the ion transfer tube was 320 °C, while the auxiliary gas was heated to 350 °C.

In multivariate statistical analysis, the metabolomics data were processed using the software metaX. Then, PCA and partial least squares discriminant analysis (PLS-DA) were employed to evaluate the overall differences between the two groups. Differential expression metabolites (DEMs) were screened based on the variable importance (VIP) in projection scores derived from orthogonal partial least squares discriminant analysis (OPLS-DA) and the FC values obtained from univariate analysis. Significantly altered variables were defined and further identified by VIP > 1 and FC > 1.2. DEMs were enriched and analyzed using MetaboAnalyst 5.0 (http//www.metaboanalyst.ca, accessed on 1 July 2024). Metabolic pathways were considered significantly enriched when the *p*-value of the metabolic pathway was <0.05.

### 2.13. Joint Analysis of Metabolomics and Proteomics

Initially, the correlations between metabolomics and proteomics were analyzed using DEMs and DEPs between the sham and CMD groups. The DEPs and DEMs were then submitted to MetaboAnalyst 5.0 for joint pathway analysis.

### 2.14. Statistics

All the statistical analyses were performed using R programming with corresponding packages available. *p*-value ≤ 0.05 (or FDR = 5% for multiple hypothesis testing) was used to define significance.

## 3. Results

### 3.1. Coronary Flow Reserve Evaluation Modeling Success

The two groups of rats were subjected to coronary blood flow measurements utilizing Doppler echocardiography. The results showed that the CMD group was significantly lower than the sham, with a CFR < 2, These results suggest that the CMD rat model was successfully constructed (Figure 1).

### 3.2. Echocardiographic Evaluation of Cardiac Function

Echocardiographic tests were performed on rats in each group. Relative to the sham group, the CMD group exhibited a significant increase in left ventricular end-systolic volume (LVESV), left ventricular end-systolic diameter (LVIDs), and left ventricular mass (LV Mass). Conversely, there was a notable decrease in end-systolic left ventricular posterior wall thickness (LVPWs), left ventricular ejection fraction (EF), and left ventricular fractional shortening (FS) in the CMD group. These findings indicate compromised myocardial contractility and diminished cardiac function in the CMD rats (Figure 2).

### 3.3. Changes in Myocardial Injury in CMD Rats

Pathologic changes in myocardial tissue were detected by HE. The findings indicated that the myocardial tissue and microvessels of rats in the sham group exhibited no significant damage. In contrast, rats in the CMD group had disturbed myocardial fiber arrangement, cardiomyocyte lysis and inflammatory cell infiltration. Thrombus and embolic microspheres were visible in the microvessels (Figure 3A). Transmission electron microscopy was employed to investigate the ultrastructure of myocardium in CMD rats, as can be seen in Figure 3B, the myogenic fibers in the sham group were neatly aligned, with clear and straight Z lines in the myonodes, and had intact mitochondrial structures. In contrast, the CMD group exhibited disorganized myogenic fibers, characterized by atrophy and fractures. Myofilaments were dissolved and cavitated, the Z lines of muscle segments were broken or blurred, and the mitochondria appeared sparse and structurally unclear, and some of them appeared to be ruptured and necrotic. There was obvious edema around the coronary microvessels, along with an uneven thickness of the basement membrane, and irregular luminal structures (Figure 3C). Serologic test results demonstrated the serum levels of CK-MB and cTn-T were significantly elevated in the rats from the CMD group compared with those in the sham group. This finding suggests that the left ventricle of the rats was injected with embolic microspheres resulting in persistent myocardial injury in the rats, which persisted without relief even after 4 weeks (Figure 3D,E). All of these results indicate that the function and structure of the rat heart and coronary microvasculature are pathologically altered after CMD.

### 3.4. Screening for DEPs and Their Underlying Biological Mechanisms

For a comprehensive understanding of the molecular determinants that influence the onset and progression of CMD, we performed an acetylation proteomics analysis of myocardial tissue. This analysis identified 8279 peptides, including 1029 acetylated peptides and 2125 proteins. To define proteins that were altered during normal and CMD transitions, we performed differential expression analysis for all proteins. Among these proteins, 41 proteins were down-regulated and 50 proteins were up-regulated for expression in CMD. We also performed labeling of DEPs on volcano and heat maps (Appendix A, Figure 4B,C). To further explore the functional changes, these proteins underwent functional enrichment analysis, including gene ontology (GO) and KEGG analysis. GO analysis revealed that DEPs were predominantly enriched in metabolic pathways, such as the fatty acid metabolic process, the purine nucleotide metabolic process, the organic acid catabolic process, the carboxylic acid metabolic process, and mitochondrial ATP synthesis coupled electron transport. These findings suggest that mitochondria may play a significant role in the pathogenesis of CMD (Figure 4D). Through KEGG pathway analysis, it was determined that pathways related to Salmonella infection, starch and sucrose metabolism, carbon metabolism, oxidative phosphorylation, and cysteine and methionine metabolism play pivotal roles in the pathogenesis of CMD (Figure 4E).

### 3.5. Construction of Co-Expression Network and Identification of Key Modules in CMD

In order to further explore the key proteins for CMD, WGCNA was performed on myocardial tissue samples to identify the most relevant protein modules. Based on the criteria of scale independence and average connectivity, a soft-thresholding power of 10 was selected. A total of 16 modules were generated using this power, with the turquoise module containing the most proteins (284 proteins) and the light cyan module containing the fewest proteins (33 proteins, Figure 5A). Furthermore, the study explored the correlation between CMD and protein modules (Figure 5B). These data indicated that the tan module exhibited the most pronounced negative correlation with CMD, encompassing 55 genes (r = −1, *p* = 3 × 10^−7^). Furthermore, a robust association was observed between module membership and protein importance within the tan modules (Figure 5C). Thus, we identified 55 genes in the tan module as proteins significantly associated with CMD. In addition, we also intersected the DEPs with the key proteins (Figure 5D). This intersection yielded 53 proteins, which were subsequently subjected to further analysis. Detailed data are shown in Appendix A.

### 3.6. Selection of Potential Biomarkers Using Supervised Machine-Learning Algorithms

In this study, we conducted an analysis of 53 key proteins using three distinct machine-learning methodologies: lasso regression, random forest and SVM. These methods were employed to identify potential biomarkers. Based on the importance of features, lasso regression selected 10 proteins (Figure 6A,B), SVM selected 37 proteins (Figure 6C), and random forest selected 20 proteins (Figure 6D). Ultimately, all methods converged on four overlapping proteins (Figure 6E). The four overlapping proteins screened by all methods, endoplasmic reticulum membrane protein complex 1 (Emc1), ankyrin-1 (Ank1), fibronectin 2 (Fbln2) and hemoglobin-binding protein (Hp), were further used to construct diagnostic models. The area under the ROC of the candidate biomarkers was calculated to assess their discriminatory accuracy. As shown in Figure 6F–H, all the model AUC values (1.000) were the same, indicating that the diagnostic models constructed for these proteins and metabolites exhibited perfect diagnostic efficiency. Consequently, these molecules are highly likely to be selected as valuable diagnostic markers. Further validation of the above diagnostic markers using ELISA showed that the levels of Emc1, Ank1, Fbln2, and Hp were significantly elevated in the myocardium and plasma of rats in the CMD group compared to the sham group (Figure 7A–D).

### 3.7. Untargeted Metabolomic Profiling

Considering that DEPs were enriched in metabolism-related pathways, we hypothesized that myocardial metabolites would more directly and accurately respond to CMD status than just the current methods. To test this hypothesis, we performed a metabolomics analysis. A total of 1961 metabolites were analysed, including lipids, amino acids, peptides, and analogues, benzene and derivatives, organic acids, etc. (Figure 8A). Multivariate statistical plots demonstrated that repeated samples within the same group were closely related to each other, whereas metabolites showed clear differentiation between groups, which indicates the reliability of the metabolome data (Figure 8B–D). A total of 410 DEMs (39 down-regulated and 371 up-regulated) were screened using a threshold VIP > 1.0, FC > 1.2 (Appendix A, Figure 8E,F). Pathway analysis of DEMs revealed significant enrichment in purine metabolism, fructose and mannose metabolism, pyruvate metabolism pathway (Figure 8G).

### 3.8. Integrated Analysis of Proteomics and Metabolomics

We next performed an integrative analysis to evaluate the relationship between DEMs and DEPs. Initially, we examined the correlation between the expression of DEMs and DEPs, specifically highlighting the correlation between the top 30 proteins and metabolites (Figure 9A). Following this, we then integrated DEM and DEP data into a combined pathway analysis, which led to the identification of several important pathways that play important roles in the pathophysiological processes of CMD, including starch and sucrose metabolism, riboflavin metabolism, pantothenate and CoA biosynthesis, citrate cycle (TCA cycle, Figure 9B). These metabolic pathways include four metabolites and six proteins, namely glucose 6-phosphate, flavin mononucleotide, adenosine 3′, 5′-diphosphate, malic acid, ectonucleotide pyrophosphatase phosphodiesterase 1 (Enpp1), 1, 4-alpha-glucan branching enzyme 1 (Gbe1), myophosphorylase (Pygm), branched chain amino acid transaminase 2 (Bcat2), succinate dehydrogenase complex flavoprotein subunit A (Sdha) and malate dehydrogenase 2 (Mdh2).

## 4. Discussion

CMD represents a significant mechanism contributing to myocardial ischemia, which critically impacts the survival and prognosis of patients with coronary heart disease [14]. The primary challenges in CMD research stem from the intricate nature of its etiology, the presence of microarteries that are not visible with imaging, and the difficulty in evaluating the effects of clinical interventions [15,16]. Consequently, early detection, diagnosis and intervention of CMD is, therefore, challenging and essential. The development of non-invasive and reliable diagnostic markers is essential for establishing cost-effective screening methods, which can substantially enhance the management of CMD. This study integrated protein expression profiles from CMD models and identified four protein markers (Emc1, Ank1, Fbln2, Hp) for the diagnosis of CMD using several supervised machine-learning algorithms.

In a three-generation exome sequencing study of congenital heart disease, mutations in Emc1 have been associated with cardiac disease, primarily aortic outflow tract abnormalities [17,18,19]. Ank1 belongs to the family of anchor proteins, which are membrane-associated scaffolding proteins widely found in a variety of tissues and cell types in various parts of the body, including neurons, cardiomyocytes, skeletal muscle cells, and epithelial cells [20]. Combined with the existing studies, Ank1 disorders were found to be closely associated with cardiac arrhythmias, and the pathological mechanism is that mutations in Ank1 result in subsequent disorganization or mislocalization of ion channels and membrane transport proteins [21]. In addition, Ank1 mutations were shown to be directly associated with dilated cardiomyopathy, cardiac hypertrophy and heart failure [22]. Fbln2 is a member of the fibronectin family of proteins that was identified as a key factor in pathological remodeling after myocardial infarction, and complete deletion of Fbln2 significantly improves survival, reduces the incidence of pericardial rupture and attenuates progressive left ventricular dysfunction [23,24]. The distinctive characteristic of Fbln2 renders it a promising therapeutic target for the prevention of progressive ventricular dysfunction following myocardial infarction [25]. Hp, a prevalent plasma protein, plays a crucial role in safeguarding against hemoglobin-induced oxidative damage. Hp was identified as a significant risk factor for acute myocardial infarction and heart failure. Hp is almost as predictive as cholesterol for acute myocardial ischemia [26,27]. It is also a risk factor for heart failure [28]. Although no studies have specifically examined the relationship between the aforementioned four protein markers and CMD, these markers are highly associated with cardiovascular disease. We were the first study to explore the expression of Emc1, Ank1, Fbln2, and Hp in a rat model of CMD. Further research is required to determine whether these proteins can serve as diagnostic markers for CMD in humans.

In fact, proteins and metabolites interact. On the one hand, proteins can influence the characterization of metabolites, and on the other hand, metabolites can influence the levels of proteins through enzymatic reactions [29]. Thus, joint analysis of proteomics and metabolomics can provide a more comprehensive understanding of CMD. We first performed acetylation proteomics and identified 91 DEPs. These DEPs are collectively involved in multiple metabolic pathways. Subsequently, we attempted to utilize metabolomics technology to more directly and accurately respond to CMD status. Integrated proteomics and metabolomics revealed that they were jointly involved in several important metabolic pathways, including sucrose metabolism, riboflavin metabolism, pantothenate and CoA biosynthesis, TCA cycle.

Riboflavin (vitamin B2) is the precursor of flavin adenine dinucleotide and flavin mononucleotide. mostly located in the mitochondria, and is central to the TCA cycle. Normal myocardial contractile function is largely dependent on high levels of oxidized TCA cycle activity. Deviations in contractile and hemodynamic responses, coupled with hypoxia and diminished energy supply, lead to substantial alterations in metabolites associated with the TCA cycle [30]. Studies have demonstrated a positive correlation between TCA cycle metabolites and the incidence of cardiovascular disease [31,32,33,34]. Citrate, the initial product following acetyl coenzyme A generation from different energy sources, was associated with cardiovascular mortality. Another study indicated that ischemia-induced succinate accumulation is a universal metabolic hallmark, a phenomenon that is caused by the reversal of Sdha. Furthermore, concentrations of pyruvate, fumarate, and malate have shown significant correlations with infarct size [35]. These results all open up the possibility of TCA circulating metabolites as new prognostic biomarkers. However, a clear link between these metabolic pathways and the development of CMD still needs to be further elucidated.

Our study is subject to several limitations. Firstly, the sample size was relatively small. Secondly, the data obtained from the metabolomic analysis were not validated. Thirdly, we only tentatively identified potential biomarkers, but their specific mechanisms of action remain unclear. Fourth, the current study is only a preliminary exploration through a rat model and does not further validate our findings in human samples. Consequently, it is necessary to increase the sample size and conduct relevant molecular mechanism studies to validate our findings. Our subsequent research focuses on investigating the mechanism of action of metabolites and proteins related to the TCA cycle in the pathophysiology of CMD.

## 5. Conclusions

In summary, we identified systematic changes in the myocardial metabolome and acetylated proteome in CMD rats. We also identified four protein markers using three supervised machine-learning feature selection methods. These diagnostic proteins could be used to construct diagnostic models with high predictive values that can effectively diagnose CMD. Integrated proteomic and metabolomic analyses reveal that TCA cycle-associated metabolites and proteins may be a key link in the pathogenesis of CMD. Although it is too early to conclude that these proteins and metabolites will become diagnostic markers for the diagnosis of CMD, we believe that our findings can contribute to the development of effective diagnostic tools that could be widely used in the clinical screening of CMD after further experimental validation.

## Figures and Tables

**Figure 1 biomolecules-14-01305-f001:**
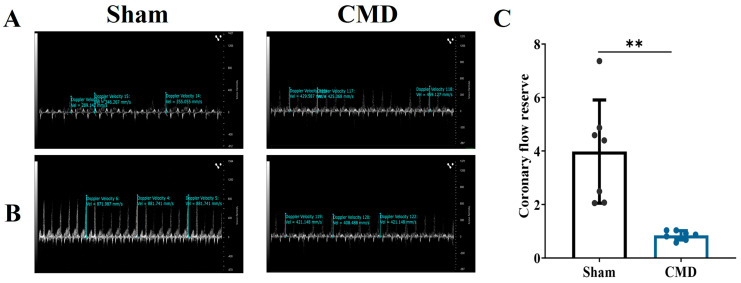
Left coronary artery pulse wave velocity imaging. (**A**) Resting state (1% isoflorane) left coronary artery trunk pulse wave image. (**B**) Congested state (2.5% isoflorane) left coronary artery trunk pulse wave image. (**C**) CFR calculated from pulse waves of experimental animals. *n* = 7 per group, ** *p* < 0.01, compared to the sham group.

**Figure 2 biomolecules-14-01305-f002:**
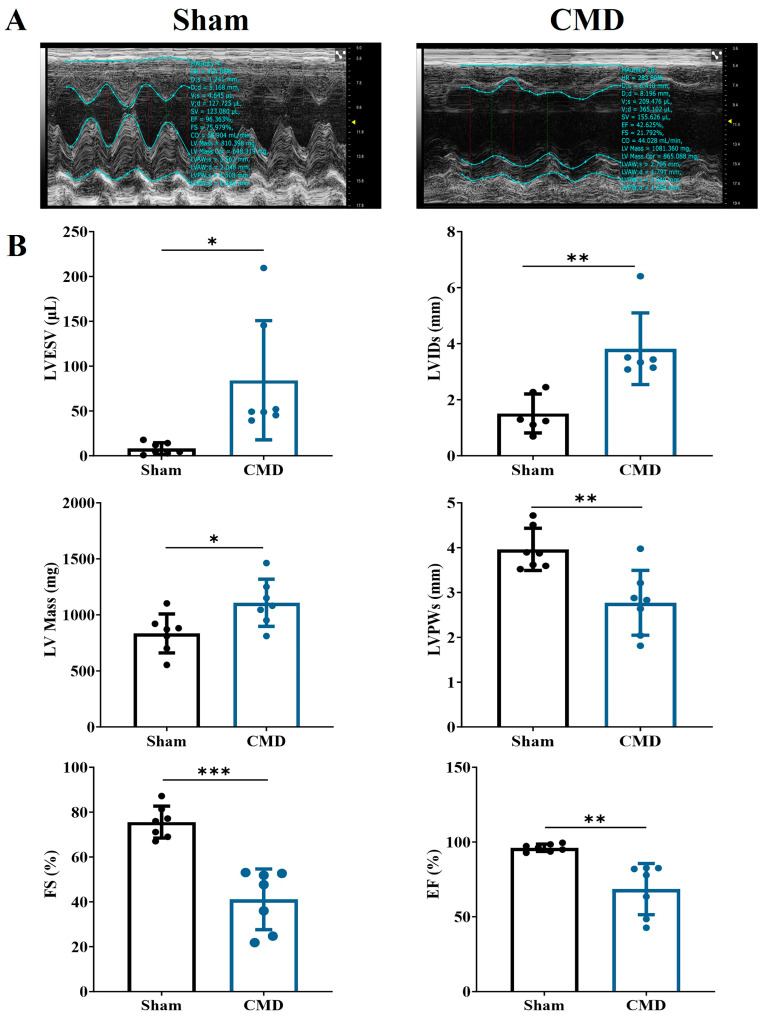
Echocardiographic examination of the rat heart. (**A**) Two-dimensional echocardiogram images. (**B**) Cardiac functional parameters. *n* = 7 per group, * *p* < 0.05, ** *p* < 0.01, *** *p* < 0.001, compared to the sham group.

**Figure 3 biomolecules-14-01305-f003:**
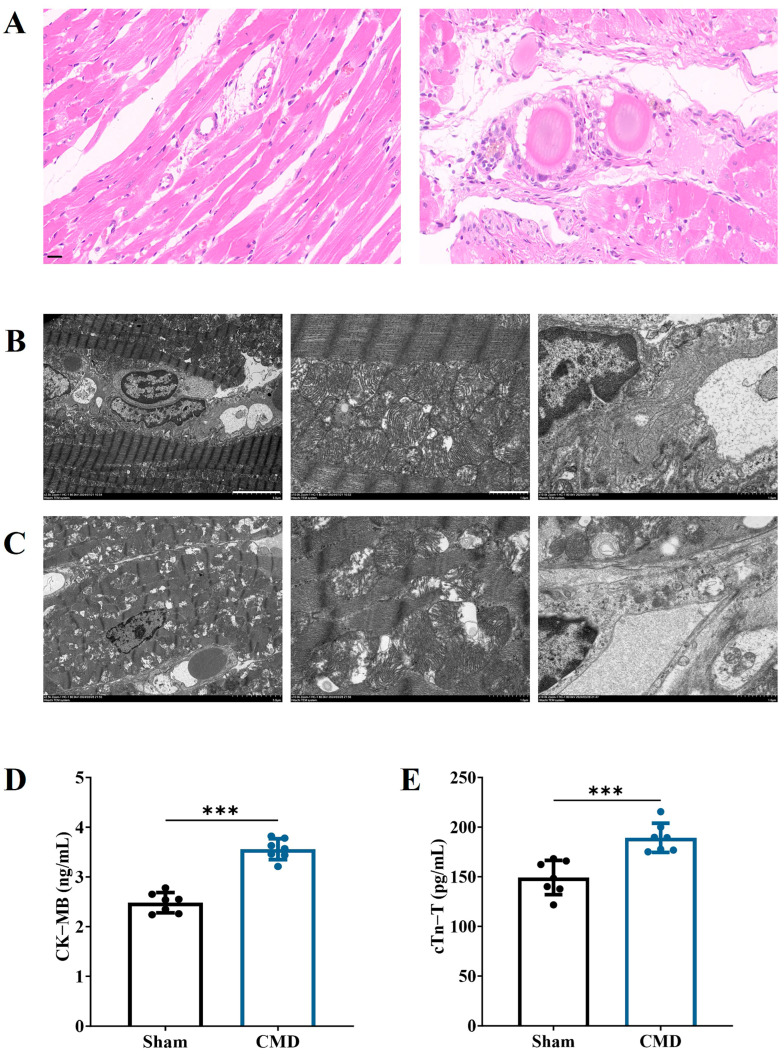
Changes in myocardial injury in CMD rats. (**A**) Representative image of HE staining of rat myocardial tissue (magnification, ×400). Scale bar: 20 μm. (**B**) Myocardial and microvascular ultrastructure of rats in sham group (magnification: ×2.5 k, ×10 k, ×10 k, respectively). Scale bar: 5 μm, 1 μm, 1 μm. (**C**) Myocardial and microvascular ultrastructure of rats in CMD group (magnification: ×2.5 k, ×10 k, ×10 k, respectively). Scale bar: 5 μm, 1 μm, 1 μm. (**D**) CK-MB. (**E**). cTnT. *n* = 7 per group, *** *p* < 0.001, compared to the sham group.

**Figure 4 biomolecules-14-01305-f004:**
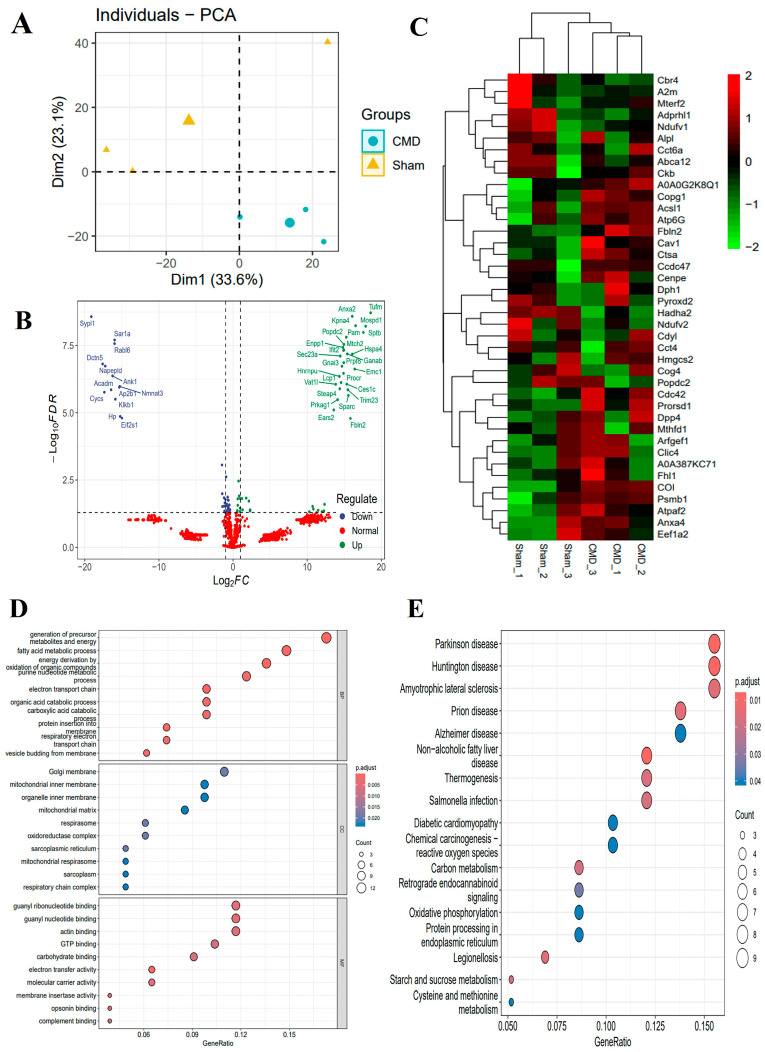
Different expression analyses and functional studies. (**A**) PCA of the identified proteins. (**B**,**C**) Volcano plots and heat maps showed the DEPs. The proteins whose expression increased in CMD are shown in green, and those whose expression decreased are shown in blue. The legend at the top right of the heat map represents log-fold changes in genes. The horizontal axis represents each sample and the vertical axis represents each protein. (**D**) GO enrichment analysis of the DEPs: biological process (BP), cellular composition (CC) and molecular function (MF). Count is the number of genes, a larger circle indicates a higher number of genes and vice versa. The x-axis indicates the ratio of the number of genes enriched to the target pathway to the total number of genes, and y-axis labels represent GO terms. (**E**) KEGG pathway annotation of the DEPs: Count is the number of genes, a larger circle indicates a higher number of genes and vice versa. The x-axis indicates the ratio of the number of genes enriched to the target pathway to the total number of genes, and y-axis labels indicate KEGG-enriched pathway.

**Figure 5 biomolecules-14-01305-f005:**
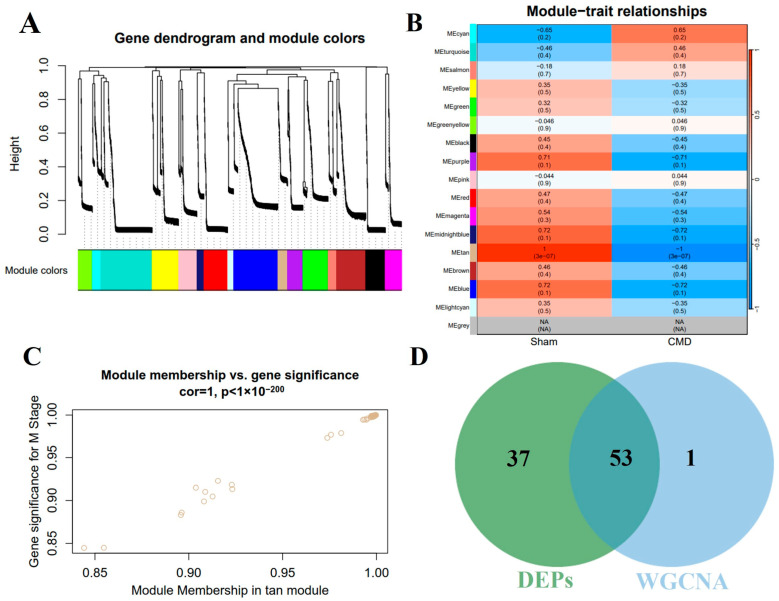
The cardiac protein co-expression network was constructed by WGCNA. (**A**) Hierarchical clustering of topological overlapping dissimilarities. (**B**) The heat map shows the relationship between module eigenvectors and CMD states. Each row corresponds to a module and each column corresponds to a trait. Each cell contains the corresponding correlation (**upper**) and *p*-value (**bottom**). (**C**) Scatter plot for correlation between protein module membership in the tan module and protein significance. (**D**) A total of 53 intersecting proteins were identified by intersecting key module proteins with DEPs via a Venn diagram.

**Figure 6 biomolecules-14-01305-f006:**
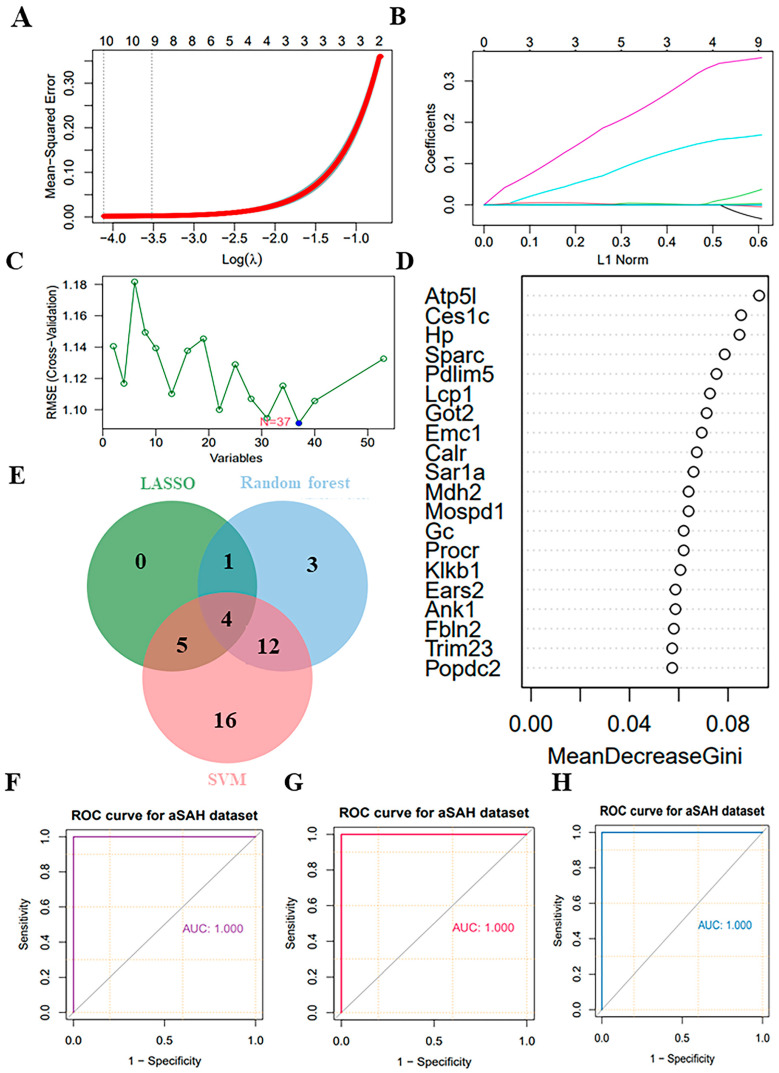
Screening candidate diagnostic markers from key proteins. (**A**,**B**) Lasso regression utilizing 53 key proteins. (**C**) SVM utilizing 53 key proteins. (**D**) Random forest utilizing 44 key proteins. (**E**) Venn diagram showed the intersection of key proteins selected by the three supervised machine-learning methods. (**F**–**H**) ROC curves for each pairwise prediction by 3 different machine-learning methods.

**Figure 7 biomolecules-14-01305-f007:**
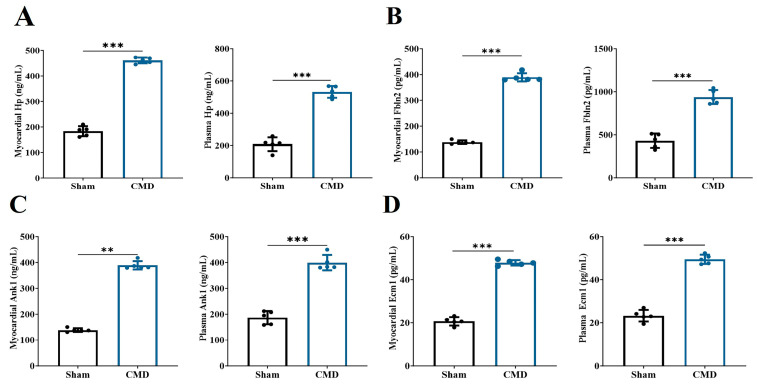
Changes in the level of diagnostic markers selected by machine learning. (**A**) Hp. (**B**) Fbln2. (**C**) Ank1. (**D**) Emc1. *n* = 5 per group, ** *p* < 0.01, *** *p* < 0.001, compared to the sham group.

**Figure 8 biomolecules-14-01305-f008:**
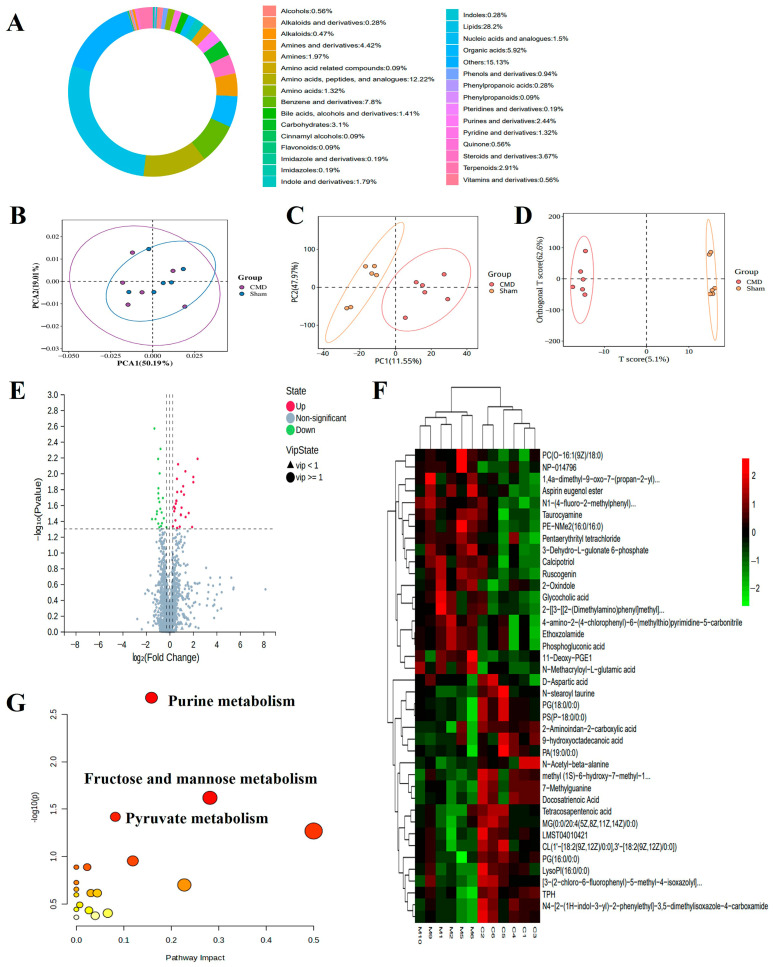
Comprehensive analysis of untargeted metabolomics in myocardial tissues of CMD and sham group rats. (**A**) The proportion of the identified metabolites in each chemical classification. Different colors indicate different metabolites classification entries, and percentages indicate the percentage of the number of metabolites in that category out of the number of all metabolites for which classification information is available. (**B**–**D**) PCA, partial least squares-discriminant analysis (PLS-DA), and OPLS-DA were performed on the identified metabolites to establish the relationship model between metabolite expression and sample category. (**E**,**F**) Volcano plots and heat maps of DEMs between CMD and sham group rats. The metabolites whose expression increased in CMD are shown in red, and those whose expression decreased are shown in green. The legend at the top right of the heat map represents log-fold changes in genes. The horizontal axis represents each sample and the vertical axis represents each metabolite. (**G**) Top significant functional pathways involved according to the DEMs. Each circle represents a metabolic pathway, the larger the circle, the greater the pathway impact.

**Figure 9 biomolecules-14-01305-f009:**
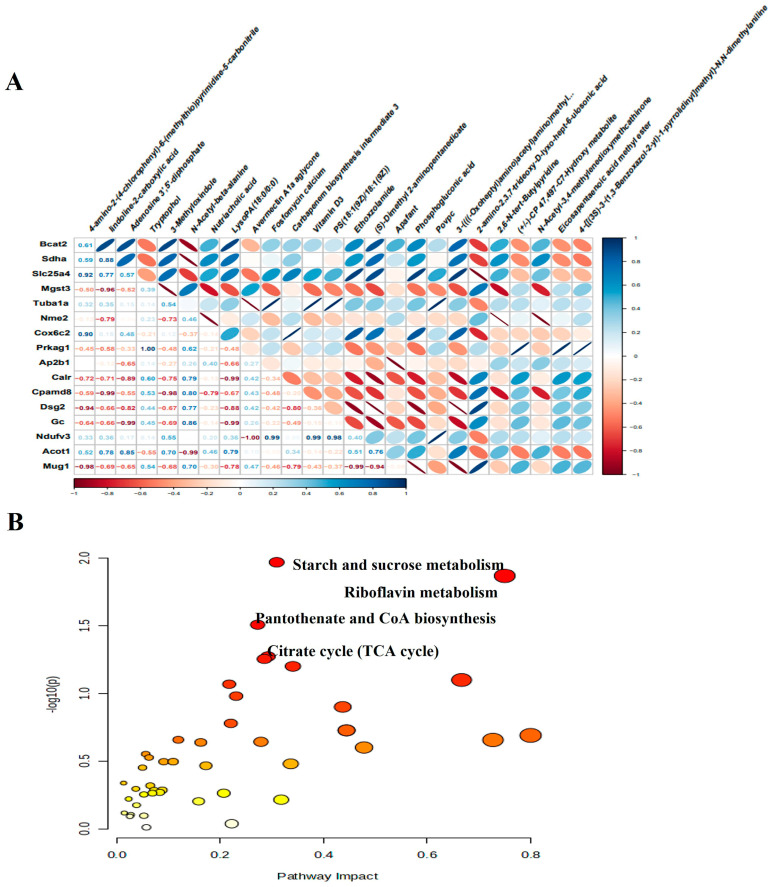
Integrated analysis of proteomics and metabolomics. (**A**) The heat map of the correlation between DEPs and DEMs. Blue represents positive correlation and is skewed to the right. Red represents a negative correlation and is skewed to the left. The darker color means the strength of the correlation. The numbers in the cells indicate the degree of correlation. (**B**) Significant functional pathways identified based on identified metabolites and proteins. Each circle represents a metabolic pathway, the larger the circle, the greater the impact of that metabolic pathway.

## Data Availability

The datasets used and/or analyzed during the current study are available from the corresponding author upon reasonable request.

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
