# Peer review of "Metabolic and Proteomic Profiling of Coronary Microvascular Dysfunction: Insights from Rat Models"

_biomolecules, 2024, doi:10.3390/biom14101305_

Round 1

Reviewer 1 Report

Comments and Suggestions for Authors

I read with interest this paper about proteomic and metabolic profiles of CMD rats as potential diagnostic markers. 

"Therefore, there is an urgent need to accurately define CMD, reveal its complex molecular features, and establish accurate detection techniques with multiple molecular markers to achieve early diagnosis and early warning, and to ensure that patients can receive accurate and effective interventions in a timely manner"

-This sentence should be referenced with recent papers about the clinical potential of multiple molecular markers in cardiovascular precision medicine (PMID: 37562936, PMID: 36794589) 

-Figure 4D-E and 5D and 8A,F and 9A. Labels are not easly readable.  

-Discussion is a bit redundant. I would suggest to start with 2-3 sentencences that summarize the main findings.

-I would suggest to comment on the necessity to validate these biomarkers in patients in order to really evaluate a clinical potential as biomarkers. 

Comments on the Quality of English Language

Minor revision is required.

Reviewer 2 Report

Comments and Suggestions for Authors

This article by Lu, Wang, and colleagues investigated the metabolic profile of coronary microvascular dysfunction (CMD) in a rat model to identify diagnostic markers. They used non-targeted metabolomics and acetylated proteomics analyses to identify 53 key proteins, with 4 proteins (Emc1, Ank1, Fbln2, Hp) determined as potential diagnostic markers for CMD.

Even though the article is interesting and informative, I have several concerns regarding some of the claims and a few recommendations.

Major

-              The title should state clearly that this study found biomarkers for CMD in a RAT model of the disease, the title right now is misleading and could make readers think this has been validated in human samples.

-              Provide all the differential expression datasets as supplementary material, it is impossible to validate anything in this article without access to the processed data. The shape of the volcano plot in Figure 4B is quite strange and I would like to see the associated data.

-              Looking for biomarkers in myocardial tissue is quite invasive, did the authors try to find any of their markers in plasma or serum? I suggest testing it in their model and repeating their diagnostic value evaluation. Would it be justified to use biopsies in patients to diagnose the disease? I am asking this as an outsider to CMD.

-              The way the “biomarkers” were selected is also troublesome. An AUC of 1 tells you that the model fits too well, which may have to do with this experiment being performed in tissue from an animal model with the same genetic background. I would recommend treating this first experiment as a “training” cohort and then adding a “validation” cohort to apply it.

-              The conclusions and limitations should state clearly that this was performed in tissue and a rat model, not humans.

-              I would strongly recommend completing the study in the animal model and doing validation in plasma/serum from human samples if they want to have any diagnostic value.

Minor

-              There is a big “chunk” of text speaking about -omics that does not really add anything interesting to the article, in 2024 most readers are familiar with these techniques and what can be achieved with them. I would suggest expanding the introduction to explain properly the molecular characteristics of CMD and why biomarkers for it are so important.

-              Line 117, is “heart transplantation” the right word? I think the authors are just harvesting the heart for their experiment.

-              In Figures 1 and 2, the quality of the image is bad and the figure legend should describe better the panels. Figures must be auto-conclusive.

-              Figure 4, increase the resolution.

-              Figure 5, could be simplified or sent to supplementary material.

-              Figure 9 needs better resolution and explanation. Panel A is just impossible to read.

-              I would suggest rethinking the figures, having less with the most important information and not just everything that comes from following a pipeline.

Round 2

Reviewer 2 Report

Comments and Suggestions for Authors

Most of my comments have been addressed, I have nothing else to say.